# Adaptive Hyperparameter Optimization for Continual Learning Scenarios

**Rudy Semola, Julio Hurtado, Vincenzo Lomonaco, Davide Bacciu**
Department of Computer Science, University of Pisa

## Abstract

Hyperparameter selection in continual learning scenarios is a challenging and underexplored aspect, especially in practical non-stationary environments. Traditional approaches, such as grid searches with held-out validation data from all tasks, are unrealistic for building accurate lifelong learning systems. This paper aims to explore the role of hyperparameter selection in continual learning and the necessity of continually and automatically tuning them according to the complexity of the task at hand. Hence, we propose leveraging the nature of sequence task learning to improve Hyperparameter Optimization efficiency. By using the functional analysis of variance-based techniques, we identify the most crucial hyperparameters, which we aim to empirically prove that they remain consistent across tasks in the sequence. This approach, agnostic to continual scenarios and strategies, allows us to speed up hyperparameters optimization continually across tasks. We believe that our future findings can contribute to the advancement of continual learning methodologies towards more efficient, robust and adaptable models for real-world applications.

## 1  Introduction

Hyperparameters play a critical role in the success of machine learning algorithms, allowing practitioners to train or fine-tune models and optimize performance [8]. Unlike model parameters, which are learned from data during the training process, hyperparameters are set before training begins and determine the model's architecture and optimization settings. Traditional hyperparameter optimization (HPO) solutions are typically designed to operate under the assumption of independent and identically distributed samples obtained all at once. However, it is often difficult to meet these assumptions in real-world environments, especially when the training data arrives incrementally.

Continual learning (CL) [22], which facilitates the progressive training of machine learning models on dynamic and non-stationary data streams [15], holds promise for enabling companies innovative trend towards Continual Learning [26, 11]. However, deploying continual learning to real-world applications remains challenging. Crucially, the role of hyperparameters in continual learning significantly impacts model performance, affecting its ability to generalize and adapt to new tasks while preventing catastrophic forgetting. Selecting, adapting, and optimizing hyperparameters are vital to retaining relevant information from past tasks while efficiently acquiring new knowledge. Nevertheless, the proper selection of hyperparameters is poorly investigated for continual strategies. These last are in many cases found via a grid search, using held-out validation data from all tasks [5] but are unrealistic for building accurate lifelong learning machines [10].

In this research, we explore relatively uncharted territory with limited prior works, delving into unexplored hypotheses that require further validation. Our effort is focused on addressing the following **research questions**, which we categorize into the dimensions of performance, efficiency, and robustness. **(RQ1)** How important is each of the hyperparameters, and how do their values affect performance in sequence learning? Which hyperparameter interactions matter? How do the values of the hyperparameters differ across different incremental learning steps? How do the answers to these questions depend on the tasks similarity of the sequence under consideration? **(RQ2)** Given that conducting HPO on each individual experience leads to improved performance but comes at the expense of computational resources, how can we enhance efficiency by leveraging hyperparameter importance information? **(RQ3)** What is the effect of HPO for each model update in relation to its robustness to the sequence order?

A primary objective of this paper is to delve into the pivotal role of hyperparameters optimization in the context of continual learning and examine possible strategies for their effective management. To summarize, we expect that **our**

**main contributions** should be:

1. Introducing and defining the problem of optimising hyperparameters through the learning experience with a realistic evaluation protocol, thereby enabling comprehensive study in this field.

2. To quantitatively assess the significance of individual hyperparameters and their interactions within each task, we propose to use the functional analysis of the variance (*fANOVA*) technique, outlined in [12, 28], for a sequence of learning tasks. The expected result should provide a quantitative basis to focus efforts on automated adaptive hyperparameter optimization for continual learning strategies.

3. To present a novel hyperparameter update rule that significantly enhances the rapid adaptation process over streaming data. Our approach introduces an adaptive policy that automatically selects task-conditioned hyperparameters on a per-step basis, thereby greatly improving the optimization effectiveness.

4. Through experiments on well-established benchmarks and baseline, we want to explore and compare our results with significant improvements in terms of average accuracy and computational cost (performance-efficiency trade-off).

5. To examine our proposal solution with existing baseline approaches in terms of robustness to the sequence order. The expected result should provide an improvement of this dimension.

## 2 Background

### 2.1 Hyperparameter Optimization (HPO)

Hyperparameter optimization (HPO) is a critical process in machine learning and statistical modelling, aimed at finding the optimal values for hyperparameters that govern the behaviour of a learning algorithm [8]. The goal is to find the hyperparameter configuration that maximizes the model's performance on a validation set or minimizes a chosen objective function.

Given a learning algorithm $\mathcal{A}$ with a set of hyperparameters $\mathcal{H}$ and a performance metric $\mathcal{M}$, the hyperparameter optimization problem aims to find the optimal hyperparameter configuration $\mathbf{h}^* \in \mathcal{H}$ that maximizes the performance metric $\mathcal{M}$ on a validation set. Formally, the problem can be defined as:

$$\mathbf{h}^* = \arg\max_{\mathbf{h}\in\mathcal{H}} \mathcal{M}(\mathcal{A}(\mathcal{D}_{\text{tr}}, \mathbf{h}), \mathcal{D}_{\text{val}}) \qquad (1)$$

where $\mathcal{D}_{\text{tr}}$ is the training dataset, $\mathcal{A}(\mathcal{D}_{\text{tr}}, \mathbf{h})$ represents the trained model obtained by applying algorithm $\mathcal{A}$ on the

training dataset with hyperparameter configuration $\mathbf{h}$, and $\mathcal{M}(\cdot)$ measures the performance of the model on the validation dataset $\mathcal{D}_{\text{val}}$.

#### 2.1.1 Hyperparameter Importance

Past knowledge about hyperparameter importance is mainly based on a combination of intuition, own experience and folklore knowledge. However, a more data-driven and quantitative approach has been introduced through the functional ANOVA framework (*fANOVA*), as presented in the work by Hutter et al. [12]. This framework enables the analysis of hyperparameter importance, offering valuable insights into the factors influencing performance.

The fundamental idea underlying hyperparameter importance lies in the extent to which a particular hyperparameter contributes to the variance in performance. Hyperparameters that significantly affect performance outcomes require careful tuning to achieve optimal results. Conversely, hyperparameters with minimal impact are considered less crucial and may warrant less attention during tuning. Functional ANOVA not only attributes variance to individual hyperparameters but also explores the interaction effects among sets of hyperparameters. This analysis sheds light on which hyperparameters can be tuned independently and which exhibit interdependence, necessitating joint tuning for optimal performance.

The study by Hutter et al. [12] applies functional ANOVA to the outcomes of a single hyperparameter optimization procedure conducted on a single dataset. Building upon this framework, Van Rijn et al. [28] extended the concept of hyperparameter importance to assess the general importance of hyperparameters across multiple datasets, providing broader insights into their significance.

### 2.2 Continual Learning (CL)

Continual Learning strategies are efficient incremental training strategies [26, 5]. An alternative approach would be to fine-tune the previous model on the new sequence of data. However, this could lead to so-called catastrophic forgetting, where performance on older data deteriorates while training on new data. The incremental learner has two goals to address this issue: to effectively learn the current task (plasticity) while retaining performance on all previous tasks (stability). Continual learning solutions have to strike a compromise between these two extremes.

Following the definitions in [5], the goal of Continual Learning is to train a model over a – possibly infinite – sequence of tasks or experiences. Each task can be represented as dataset $D(x^{(t)}, y^{(t)})$ sample from a distribution $D^{(t)}$ in time $t$, were $x^{(t)}$ is the set of samples for task $t$, and $y^{(t)}$ the corresponding label. As a result, the learner function should be able to model the cumulative probabil-

ity distribution of the data $\mathcal{P}(Y|X)$ where $X$ is the set of all samples for all tasks and $Y$, are their corresponding labels. Since this probability distribution is intractable, continual strategies perform indirect optimization by minimizing the following:

$$\sum_{t=1}^{\mathcal{T}} \mathbb{E}_{x^{(t)}, y^{(t)}} [\mathcal{L}(\mathcal{A}_t^{CL}(x^{(t)}; \theta), y^{(t)})] \qquad (2)$$

with limit or no access to previous data $(x^{(t')}, y^{(t')})$ when training tasks $t > t'$. Then, Continual Learning methods seek to optimize the parameters $\theta$ by minimizing the loss expectancy for all tasks in the sequence $\mathcal{T}$.

## 2.3 HPO in Continual Learning Scenarios

### 2.3.1 Problem Formulation

In continual learning, models are trained and applied for prediction without having all training data beforehand. The process that generates the data may change over time, leading to concept drift, a usually unpredictable shift over time in the underlying distribution of the data. It is crucial to understand the dynamics of concept drift and its effect on the search strategy used by the HPO technique in order to design a successful strategy in a non-stationary sequence data stream. Considering the equation 1 designed for stationary environments, HPO for a particular task $t$ in sequence learning $\mathcal{T} = t_0, ..., t_M = \{t\}$ can be described as the following optimization problem:

$$\mathbf{h}_t^* = \arg\max_{\mathbf{h} \in \mathcal{H}} \mathcal{M}(\mathcal{A}_t^{CL}(\mathcal{D}_{\text{tr}}^{(t)}, \mathbf{h}), \mathcal{D}_{\text{val}}) \qquad (3)$$

Here $\mathcal{D}_{\text{val}}$ is the subset of the validation set for the current task and previous tasks and $\mathcal{A}_t^{CL}$ is the continual learner achieved so far with hyperparameter configuration $\mathbf{h}$ sought only on currently available dataset $\mathcal{D}_{\text{tr}}^{(t)}$.

### 2.3.2 Related Works

Hyperparameters play a significant role in continual learning by influencing the model's ability to adapt to new tasks while retaining knowledge from previous tasks. Note that strategies tackling the continual learning problem typically involve extra hyperparameters to balance the stability plasticity trade-off. These hyperparameters are in many cases found via a grid search, using held-out validation data from all tasks [5]. In particular, the typical assumption is to have access to the entire stream at the end of training for model selection purposes. After that, as a common practice, the hyperparameters are set to a fixed value throughout all incremental learning sessions. One simple motivation is that tuning hyperparameters is a major burden and is no different in the continual setting. Another motivation is the simplicity of implementation. Nevertheless, this inherently

violates the main assumption in continual learning, namely no access to previous task data and in practical scenarios is unrealistic for building accurate lifelong learning machines [10, 4].

In the following, we present a comprehensive summary of related works concerning adaptive HPO and CL, categorizing them into three classes.

**Dynamic Task-Specific Adaptation.** Real-world learning presents a dynamic environment where the optimal hyperparameter configuration may evolve over time. In response to this challenge, recent work by Gok et al. [9] focuses on continual learning and investigates the necessity of adaptive regularization in Class-Incremental Learning. This approach dynamically adjusts the regularization strength based on the specific learning task, avoiding the unrealistic assumption of a fixed regularization strength throughout the learning process. Empirical evidence from their experiments highlights the significance of adaptive regularization in achieving enhanced performance in visual incremental learning. Concurrently, Wistuba et al. [29] address the challenge of practical HPO for continual learning. They propose adjusting hyperparameters such as learning rates, regularization strengths, or architectural choices for different tasks, allowing the model to adapt its behaviour to each task's specific requirements. Their empirical findings demonstrate improvements in performance through this approach. However, it is worth noting that despite the demonstrated performance gains, Wistuba et al. [29] do not fully leverage the inherent nature of continual learning problems or exploit the potential knowledge transfer from previous HPOs. Incorporating transfer learning techniques and capitalizing on insights from prior tasks could further enhance the model's performance and overall efficiency in continual learning scenarios.

**Transfer Learning and Knowledge Distillation.** [27] has explored the idea of adapting hyperparameters to optimize performance for individual tasks, facilitating automatic knowledge transfer from previous HPO endeavours across datasets. Furthermore, hyperparameters and dynamical architecture chance can facilitate transfer learning, where knowledge and hyperparameters learned from previous tasks are leveraged to improve performance on new tasks. Knowledge distillation techniques can be applied [25, 24], where hyperparameters guide the transfer of knowledge from a larger or more accurate model to a smaller or more specialized model.

**Empirical Studies.** Hyperparameters can govern the use of memory replay or experience replay mechanisms to mitigate catastrophic forgetting. The hyperparameters determine aspects such as the importance of past data samples, the frequency of replay, or the balance between old and new data, thereby influencing the impact of replay on model performance as highlighted in [21, 5, 10].

# 3 Adaptive Hyperparameter Optimization for Continual Learning Scenarios

## 3.1 Methodology and Key Assumptions

We hypothesize that the assumption of hyperparameters constancy in all sequence stream data is unrobust and ineffective in continual learning settings. Firstly, the common practice of using held-out validation data from all tasks inherently violates the main assumption of having no access to previous task data. Secondly, the assumption of fixed hyperparameters in all sequence stream data is unrealistic for building effective continual learning real-world systems. Finally, assuming to work on more practical scenarios, optimizing hyperparameters over the entire data sequence is not possible and real solutions involve changing or not the hyperparameter when it detects a distribution shift or model's performance decay.

---

**Algorithm 1** Adaptive Hyperparameters Tuning for Continual Learning

---

**Require:** $\mathcal{H} = \{H_n\}$ configuration space with hyperparameters $N$; $\mathcal{T} = \{t\}$ sequence tasks; $\mathcal{A}^{CL}$ continual learner

1: **Initialization**
2: **for** $(t, i)$ **in** $\mathcal{T}$ **do**
3:     **if** $i < m$ **then**              ▷ First $m$ tasks
4:         $h_t^* = \text{hpo}(\mathcal{H}, t, \mathcal{A}_t^{CL})$
5:         $\{\mathcal{H}_n, \text{v}\} = \text{get\_param\_imp}(f\text{ANOVA}, \text{hpo}, \mathcal{H})$
6:         $\mathcal{H}^k = \text{top\_k\_hp}(\{\mathcal{H}_n, \text{v}\}, k)$
7:     **else**              ▷ Rest of other tasks
8:         $h_t^* = \text{hpo\_warm\_start}(\mathcal{H}^k, h_{t-1}^*, t, \mathcal{A}_t^{CL})$
9:     **end if**
10: **end for**
11: **return** $h_{\mathcal{T}}^*$ best configuration in all the sequence tasks

---

The idea behind Adaptive Hyperparameters Tuning for Continual Learning (Algorithm 1) is simple. In the first $m$ tasks, we select the best configuration in $\mathcal{H}$ as in a typically stationary setting. Exploiting the *fANOVA* evaluator, we compute parameter importances based on completed trials in the given HPO and associated continual learner $\mathcal{A}_t^{CL}$. In the remaining tasks, we speed up the optimization process based on the importance of each parameter. The proposal method automatically selects the $k$ most important parameters to be changed and keeps fixed the others with the optimal value computed in the previous task. Note that this policy selection of the parameters to tune is automatic and agnostic to CL strategies and sequence tasks (and their order).

*hpo*: Implement a specific HPO for each model update, i.e. grid search, population-based training or Bayesian optimization. The latter should speed up the HPO if we start from the best configuration in the previous task. For this reason, we want to use Bayesian Optimization and Hyper-

band (BOHB) [7] that performs robust and efficient hyperparameter optimization at scale by combining the speed of Hyperband searches with the guidance and guarantees of convergence of Bayesian Optimization among tasks.

*get\_param\_imp*: Evaluate parameter importances based on *fANOVA* evaluator in the given HPO and $\mathcal{A}_t^{CL}$. The function returns the parameter importances as a dictionary $\{\mathcal{H}_n, \text{v}\}$ where the keys consist of specific parameter $H_n$ and values importance $\text{v} \in \{0, 1\}$.

*top\_k\_hp*: function that return $\mathcal{H}^k$ as a subspace of $\mathcal{H}$ after ordering the parameters by importance.

*hpo\_warm\_start*: the idea is to speed up the tuning process among the sequence tasks by starting from the optimal parameters found in the previous task $h_{t-1}^*$. Furthermore, knowing the importance of each parameter we select the subspace of $\mathcal{H}$ with the most $k$ important to be tuned and keep fixed the others.

We believe that the tasks in the sequence, even if they have different distributions, have enough task similarity to be exploited in the sequence HPO to efficiently tune the parameters on the current task. To do this, our idea is to exploit the hyperparameter importance information to automatically select the parameters to be tuned and which could be fixed dynamically in the non-stationary learning sequence.

## 3.2 Experimental protocol

The goal of this section is to describe the protocol, benchmarks, baseline, and continual strategies that we plan to use in the experiments.

### 3.2.1 Benchmarks and Metric

Continual learning algorithms are evaluated by *benchmarks*: they specify how the stream of data is created by defining the originating dataset(s), the number of samples, the criteria to split the data into different tasks and so on. In literature, different benchmarks are used to evaluate results.

In this paper, we would conduct experiments using benchmarks from two distinct scenarios: Class Incremental, where each task introduces new classes without revisiting old ones in the training stream, and Domain Incremental, where each task presents new instances for existing classes without reusing old instances in the training stream [15]. We have selected three for **Class Incremental Learning**: Split-CIFAR10 [30], Split-TinyImageNet [20] and CORe50-NC [18]. These benchmarks are derived respectively from CIFAR-10 [13] and TinyImagenet [14] datasets while CORe50-NC is a benchmark specifically designed for continual learning. For **Domain Incremental Learning** we have selected Rotated-MINIST derived from the MNIST dataset [6] and CORe50-NI.

Each dataset exhibits visual classification learning. We will train all the models with a minimum of 10 tasks. We will resort to the standard metrics for evaluation, i.e. accuracy, which measures the final performance averaged over all tasks (also defined as *stream accuracy*, SA) [15]. In both incremental scenarios, higher values indicate better performance.

### 3.2.2 Baselines

We intend to compare our solution with two opposite approaches used both in literature [29] and more practical scenarios for hyperparameter optimization for sequence task learning.

- The upper bound in terms of performance and lower bound for computational cost is HPO for each model update as used in [29]. We plan to use grid search or Bayesian optimization as an HPO technique to improve as possible the performance comparison. The computational cost expected to achieve is $|\mathcal{H}| * |\mathcal{T}|$ which is the worst case.

- The lower bound in terms of performance and upper bound for computational cost is to perform HPO only in the first experience and keep fixed the configuration for the rest of the sequence learning. The computational cost expected to achieve is $|\mathcal{H}|$ which is the best case.

### 3.2.3 Continual Learning strategies

We selected five strategies among the most popular and promising rehearsal and regularization approaches.

In particular, for rehearsal, we intend to use the following.

**Experience Replay (ER).** We selected Replay [3, 21] because it is an effective continual strategy for practical scenarios. In particular, it is a simple way to prevent catastrophic forgetting, and it performs better with respect to more complicated strategies. In our future experiments, we plan to explore different buffers with different policies to select-discard samples.

**Greedy Sampler and Dumb Learner (GDumb)** [23] is a simple approach that is surprisingly effective. Compared to other rehearsal methods, with the same memory size, this strategy is more efficient, in terms of execution time and resources. In a particular setting, this simple strategy can outperform other approaches.

**Dark Experience Replay (DER/DER++)** [2], as a more recent replay method, relies on dark knowledge for distilling past experiences, sampled over the entire training trajectory. With respect to ER, DER converges to flatter minima and achieves better model calibration at the cost of limited memory and training time overhead.

For well-established regularization methods, we intend to employ the following category of continual learners because we believe they are more sensitive to hyperparameter selection and necessitate hyperparameters that adapt dynamically throughout the learning sequence.

**Learning-without-Forgetting (LwF)** [16] is a knowledge-distillation approach where the teacher branch is the model from the previous task, and the student branch is the current model. The aim is to match the activations of the teacher and student branches, either at the feature or logit layer.

**Synaptic Intelligence (SI).** [30] introduces intelligent synapses that bring some of this biological complexity into artificial neural networks. Each synapse accumulates task-relevant information over time and exploits this information to rapidly store new memories without forgetting old ones.

Table 1: The continual strategies with their specific and general hyperparameters. $mem\_size$: replay buffer size; (*) both the optimizer and model could have further parameters.

| Strategy | Hyperparameters |
|---|---|
| ER (replay) | $mem\_size$ 
 *buffer type* 
 *storage policy* |
| GDumb | $mem\_size$ |
| DER/DER++ | $mem\_size$ 
 *alpha* 
 $beta = 0$ (DER) 
 *with* $beta \neq 0$ (DER++) |
| LwF | *alpha* 
 *temperature* |
| SI | *lambda* 
 *eps* |
| General Hyperparameters | *optimizer\** 
 *lr* 
 *training epochs* 
 *batch size* 
 *model\** |

We report in Table 1 the selected ad-priori most interesting hyperparameters for each continual method and additional hyperparameters less specific to the learner. This selection for the experiments has a twofold intention. Primarily, we aim to spotlight parameters specific to individual continual learners, particularly pertinent to addressing RQ1. Additionally, we recognize the significance of general parameters like learning rate, which should impact overall performance. Therefore, we intend to showcase their inclusion as a demonstration of the adaptability of our solution across a wide array of incremental scenarios and strategy types.

### 3.2.4 Implementation details

We will run experiments on three different seeds and report their average. For each benchmark, the evaluation protocol will be split by pattern. First, we will split the overall dataset into 90% model selection and 10% model assessment patterns. Then, we will use 15% of model section data as a validation set and a batch size of 32 examples. For the experimental part, we intend to use Avalanche [19] the reference continual learning framework based on PyTorch and Ray Tune for the hyperparameter optimization [17]. To quantitatively assess the significance of individual hyperparameters and their interactions within each task, we will leverage various metrics, as outlined in [12, 28], and will implement them within the Optuna framework [1].

Finally, for the analysis of the robustness with respect to sequence order, we will conduct experiments across three distinct orders on the benchmarks defined in Subsection 3.2.1 and subsequently report their average with associated standard deviations. We intend to employ the stream accuracy metric for each model update, the outcomes of which will be visually presented in plots that facilitate a comparative analysis of our solution against all baseline methods.

## 4 Expected Results and Discussion

Our objective is to empirically demonstrate that:

- The hypothesis of automatically adjusting the most effective continuous parameters for each update can significantly expedite the accuracy of the overall sequence task.

- The quite task-independent hyperparameters can be computed from the initial tasks in a similar sequence of streamed data and can considerably accelerate the hyperparameter processes in practical scenarios while maintaining an advantageous performance-efficiency tradeoff.

- By striking a favourable balance between performance and efficiency, we hypothesize that our proposal method will greatly enhance robustness, particularly in terms of sequence order.

### 4.1 Important and Adaptive Hyperparameters for CL Analyses

**(RQ1)** How important is each of the hyperparameters, and how do their values affect performance in sequence learning? Which hyperparameter interactions matter? How do the values of the hyperparameters differ across different incremental learning steps? How do the answers to these questions depend on the task similarity of the sequence under consideration?

We want to demonstrate that performance variability is often largely caused by a few hyperparameters that define a subspace to which we can restrict configuration. Moreover, given a specific continual learner and sequence of similar tasks, we believe that this small set of hyperparameters responsible for the most variation in performance is the same set of hyperparameters across the sequence.

### 4.2 Performance-efficiency Analyses

**(RQ2)** Given that conducting HPO on each individual experience leads to improved performance but comes at the expense of computational resources, how can we enhance efficiency by leveraging hyperparameter importance information?

According to similar work like [28], we want to demonstrate empirically that the hyperparameters determined as the most important ones indeed are the most important ones to optimize also in sequence task learning. In particular, given a continual learner, only a small set of hyperparameters are responsible for most variation in performance and this is the same for all tasks in a sequence. As a result, we only perform hyperparameter optimization in $\mathcal{H}$ for the initial $m$ tasks with $m < |\mathcal{T}|$ and speed up the hyperparameter optimization through the remaining data stream.

### 4.3 Robustness Analyses

**(RQ3)** What is the effect of HPO for each model update on robustness to the sequence order?

Based on the insights from [29], we intend to conduct further experiments to emphasize the importance of performing HPO for each model update and emphasize its robustness concerning the sequence order. We anticipate observing enhanced performance trends across the stream and reduced variance.

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
