# OpenReview forum: "Adaptive Hyperparameter Optimization for Continual Learning Scenarios"
_continualai.org/CLAI/2023/Unconference_Preregistration_Track — 1st CLAI Unconf_

### Official Review · Reviewer_eHmd · 2023-08-07
**Well articulated research questions for adaptively choosing hyperparameters during continual learning**

**Clarity:** 3
**Originality:** 4
**Soundness:** 3
**Significance:** 3
**Rating:** 8
**Confidence:** 5

**Review:**

This work aims to study hyperparameter optimization for continual learning. This is very important as many methods achieve state-of-the-art performance by doing a grid search or ad hoc search over hyperparameters, which often isn't even reported. In addition to studying the role of hyperparameter selection, they aim to automatically select hyperparameters during continual learning. While some aspects of the proposal are unclear, they have well articulated research questions.

**Strengths:**

They have identified an important problem in continual learning, as it is unrealistic for hyper-parameters to be chosen with something like a grid search for continual learning, where an algorithm should be presented with a sequential learning problem from an unknown distribution. They have well defined research questions and a relatively good protocol.

**Weaknesses:**

It would make sense to provide a motivating study of one or more continual learners and demonstrate their sensitivity to hyperparameters. Many of the ones that systems are most sensitive to, e.g., the amount of pre-training or the batch size, can greatly impact results. While they have a very general method for hyperparameter optimization, it would be good to choose methods with relevant hyperparameters that they will use with their method, which I assume are just related to the optimizer, but there are often many algorithm specific hyperparameters in continual learning.

**Questions:**

Why weren't more recent continual learning methods selected? I think validating the method using some general method, e.g., experience replay, and then showing that it also can be applied to some more advanced methods would provide adequate evidence.



**Protocol:**

Overall, I think the protocol makes sense for the research questions.

The baseline methods compared against I think do not make sense, except for ER and GDumb. EWC and iCARL have been shown in many papers to perform poorly compared to many methods. While they are important for historical reasons, pretty much every method beats them and has done so since they were proposed.

The upper and lower bounds for hyperparameters make sense.

I think the work would have significantly more impact if they also tested larger and higher resolution datasets, in addition to providing the computational/memory requirements needed for adaptive hyperparameter optimization for them. The method should be shown to scale.

---

### Official Review · Reviewer_vDpg · 2023-08-21
**Review for "Adaptive Hyperparameter Optimization for Continual Learning Scenarios"**

**Clarity:** 4
**Originality:** 3
**Soundness:** 3
**Significance:** 3
**Rating:** 8
**Confidence:** 4

**Review:**

The paper proposes a new framework of hyperparameter optimisation in continual learning environments aided by $f$-ANOVA. The paper builds on the importance of HPO in non-stationary environments and why it remains an under-explored topic of research. Further, motivated by the practical real world use cases of continual learning agents and the practicality of optimal hyper-parameters the paper provides an intuitive framework of adapting hyper-parameters in CL environments using $f$ANOVA to first rank hyper-parameters in their order of importance and then further find optimal values of these hyper-parameters using Bayesion optimisation and Hyperband (BOHB) HPO.

**Strengths:**

1. The paper provides a clear intuition and motivation into the importance of HPO in non-stationary environments like that of CL. The motivation is further supported by the practical reliability of optimal hyper-parameter in accurately adapting learning agents to change in distribution.
2. The paper is well written and clear. All statements and list of contributions are clearly stated in Section 1.
3. The paper provides a clear literature review and provides intuitive explanation into the limitations of prior works which have explored HPO in continual learning environments.
4. The use of BOHB is well justified and the framework demonstrated in Algorithm 1 is clear.
5. The paper provides strong justification for the choice of the experiment protocol along with expected results and discussions.

**Weaknesses:**

1. The assumption in section 3.1, last paragraph: "We believe that the tasks in the sequence, even if they have different distributions, share some structure that could be exploited in the sequence HPO among tasks." is unclear. What do the authors mean by "structure"?
2. It is also further unclear how optimising for optimal hyper-parameters for first m tasks in a sequence will be transferrable to the remaining tasks in the sequence assuming: a. Its long sequence with m << total tasks in sequence and that sequence length is unknown. b. There is progressive significant distribution shift over the length of the sequence such that there is much to no similarity of task k and task k+t. The framework proposed in the paper assumes that the parameter importance ranking obtained as a result of $f$ANOVA in the first m tasks also holds for the remainder of the sequence which is not necessarily practical and reflecting of real world setting.
3. What is unclear is how this HPO strategy will help in mitigating forgetting, it seems more biased towards improving forward transfer but not necessarily backward transfer.

**Questions:**

Refer to weaknesses section.

**Protocol:**

The evaluation protocol provided in the manuscript is thorough and apt for a 6-month timeline. To add, it would be interesting to have more online based settings rather than offline CL setting to provide real world justification.

---

### Official Review · Reviewer_a5xf · 2023-08-21
**Interesting research proposal, more details on experiments**

**Clarity:** 4
**Originality:** 3
**Soundness:** 3
**Significance:** 3
**Rating:** 7
**Confidence:** 4

**Review:**

This paper proposes to study the question of hyperparameter optimization in continual learning. First it plans to study how to optimize the hyperparameter and how sensitive it is during training due to the online and continual nature of the task. Second, it aims to propose an online update procedure to reduce the need for offline optimization of multiple experiences. Lastly, it aims to study robustness. Overall, this represents a well planned research study on hyperparameter optimization for continual learning and is of good interest to the community.

**Strengths:**

- This paper proposes three interesting research questions that are worth studying.
- This paper provides a good foundation of the motivation of the research.
- This paper provides a solid experimental protocol with a proposed algorithm.

**Weaknesses:**

- More information on the set of hyperparameters and why they might be influential.
- Hypotheses on what may or may not happen and what to conclude in each case.

**Questions:**

N/A

**Protocol:**

- The experiment protocol is good. Datasets are identified.
- It would be better if the set of hyperparameters can be identified too.
- More information on how to experiment with different task relatedness and sequential ordering.

---

### Decision · Program_Chairs · 2023-09-12

**Decision:**

Accept

**Comment:**

Dear authors,

Congratulations, your paper has been accepted at the ContinualAI Unconference 2023! We look forward to engaging in further discussions with you and others in the community.

Details will follow shortly regarding camera-ready versions. Please do take the feedback from reviews into account.

Thanks!